# ORTHOGONAL CALIBRATION FOR ASYNCHRONOUS FEDERATED LEARNING

## ABSTRACT

Asynchronous federated learning mitigates the inefficiency of conventional synchronous protocols by integrating updates as they arrive. Due to asynchrony and data heterogeneity, learning objectives at the global and local levels are inherently inconsistent—global optimization trajectories may conflict with ongoing local updates. Existing asynchronous methods simply distribute the latest global weights to clients, which can overwrite local progress and cause model drift. In this paper, we propose ORTHOFL, an orthogonal calibration framework that decouples global and local learning progress to reduce interference. In ORTHOFL, clients and the server maintain separate model weights. Upon receiving an update, the server aggregates it into the global weights via a staleness-aware moving average. For client weights, ORTHOFL computes the global weight shift during the client's delay and removes its projection onto the direction of the received update. The resulting parameters lie in a subspace orthogonal to the client update and preserve the maximal information from the global progress within the orthogonal hyperplane. The calibrated shift is then merged into the client weights for further training. Extensive experiments demonstrate ORTHOFL improves accuracy by **9.6%** and achieves a speed-up of **12×** compared to synchronous methods. Moreover, it consistently outperforms state-of-the-art asynchronous baselines under various delay patterns and heterogeneity scenarios.

## 1 INTRODUCTION

Traditional federated learning (McMahan et al., 2017) follows a *synchronous* update procedure, requiring the server to wait for all clients to complete local training before aggregating their updates. Such synchronization can become significantly inefficient when client resources (e.g., computation power, network bandwidth, data volume) are highly heterogeneous. Asynchronous federated learning addresses this inefficiency by aggregating client updates upon their arrival, thus minimizing idle time caused by slower clients. In this setting, the global model continuously evolves when clients are performing local training, making their updates potentially outdated by the time they reach the server.

Current methods (Xie et al., 2019; Liu et al., 2024a; Zang et al., 2024; Su and Li, 2022) handle staleness by applying decay weighting, buffering, or heuristic corrections to outdated updates, after which they directly distribute the updated global weights to clients for further local training. While these methods account for temporal drift, they overlook spatial drift—divergence in optimization paths due to data heterogeneity, where the global model seeks parameters that generalize across the overall distribution, while individual client models minimize loss on local data. This misalignment is further amplified by asynchrony, as stale local updates become disconnected from the current global trajectory. Consequently, overwriting client models with the latest global weights can introduce conflicting optimization directions, reverse local gains, and cause unstable training dynamics.

To address the challenge, we introduce a foundational principle: global and local objectives should be explicitly decoupled to minimize interference and accurately reflect their distinct optimization roles. We propose ORTHOFL, a novel asynchronous method that maintains separate global and local model weights and uses a geometric projection to eliminate interfering update directions. Our method is motivated by the idea that high-dimensional parameter spaces admit many viable directions for effective optimization (Wortsman et al., 2021). While some directions interfere with previously

learned knowledge, others are compatible and preserve performance. ORTHOFL leverages this property to calibrate global weight shift, removing components that would disrupt local optimization.

Specifically, ORTHOFL introduces *orthogonal calibration*, which achieves two goals: (1) eliminating interference by sharing global information orthogonal to client updates, and (2) selecting informative direction within this orthogonal hyperplane to maximize useful knowledge transfer. Once receiving a client update, the server integrates it into the global model through an adaptive moving average accounting for staleness. For the client model, ORTHOFL computes the accumulated global weight shift occurring during client delay, projects it onto the client's update direction, and subtracts this projected component. The resulting parameters lie in a subspace orthogonal to the client update and retain the maximal portion of the global shift within this subspace. Finally, the calibrated global shift is merged into the client model for subsequent training.

For evaluation, we incorporate realistic delay distributions to reflect the heterogeneous nature of real-world deployments. ORTHOFL demonstrates an average of 9.6% accuracy improvement across datasets from diverse application scenarios compared to synchronous methods and a $12\times$ speedup in reaching a target accuracy, Moreover, it outperforms state-of-the-art asynchronous baselines. We also explore various simulated delay distributions and data heterogeneity levels to understand their impact on model performance and convergence speed. In summary, our contributions are as follows:

- We identify and analyze the key challenges of asynchronous federated learning—the inconsistency of global and local objectives and the detrimental effect of staleness under heterogeneous conditions.
- We propose a novel orthogonal calibration method that maintains separate global and local model weights. It projects global shifts onto orthogonal subspaces of local updates before sharing them with clients. This approach reduces interference, preserves meaningful contributions from both global progress and local updates, and enhances knowledge sharing.
- We demonstrate the effectiveness and robustness of ORTHOFL through comprehensive experiments on multiple datasets and various delay scenarios, providing insights on practical design considerations for large-scale federated learning systems.

## 2 RELATED WORKS

We review two core directions relevant to our study. Connections with broader areas including asynchronous stochastic gradient descent and orthogonal gradient descent are discussed in Appendix G.

**Federated Learning and Heterogeneity Problem**. Federated learning (McMahan et al., 2017) is a distributed learning paradigm that allows multiple parties to jointly train machine learning models without data sharing, preserving data privacy. Despite the potential, it faces significant challenges due to heterogeneity among participating clients, which is typically classified into two main categories: data heterogeneity and system heterogeneity. Data heterogeneity appears as clients own non-IID (independent and identically distributed) data (Li et al., 2020; Karimireddy et al., 2020; Wang et al., 2020; Zhang et al., 2023c). The difference in data distribution causes the local updates to deviate from the global objective, making the aggregation of these models drift from the global optimum and deteriorating convergence. System heterogeneity refers to variations in client device capabilities, such as computational power, network bandwidth, and availability (Wang et al., 2020; Zhang et al., 2021; Li et al., 2021; Fang and Ye, 2022; Alam et al., 2022; Zhang et al., 2024a). These disparities lead to uneven progress among clients, and the overall training process is delayed by slow devices. Traditional federated learning approaches rely on synchronization for weight aggregation (McMahan et al., 2017; Li et al., 2020; Reddi et al., 2020), where the server waits for all clients selected in a round to complete and return model updates before proceeding with aggregation. This synchronization leads to inefficient resource utilization and extended training times, particularly in large-scale deployments involving hundreds or thousands of clients. Addressing the heterogeneity issues is a critical problem for improving the scalability and efficiency of federated learning systems in real-world deployment.

**Asynchronous Federated Learning**. Much of the existing literature focuses on staleness management by assigning weights for aggregating updates according to factors including delay in updates (Xie et al., 2019), divergence from the global model (Su and Li, 2022; Zang et al., 2024) and local losses (Liu et al., 2024a). For example, Xie et al. (2019) let the server aggregate client updates into the global model with a weight determined by staleness. Another line of research caches client updates at the server and reuses them to calibrate global updates (Gu et al., 2021; Wang et al.,

2024a). For example, Wang et al. (2024a) maintain the latest update for every client to estimate their contribution to the current aggregation and calibrate global updates. Furthermore, semi-asynchronous methods (Nguyen et al., 2022; Zang et al., 2024) balance between efficiency and training stability. For example, Nguyen et al. (2022) buffer a fixed number of client updates before aggregation. Zhou et al. (2024) adopt client clustering based on gradient similarity and a two-stage aggregation scheme with semi-asynchronous intra-cluster and synchronous inter-cluster updates. We select representative methods from each category for our comparisons. Besides, some works improve efficiency from a different perspective—through enhanced parallelization. Methods include decoupling local computation and communication (Avdiukhin and Kasiviswanathan, 2021) and parallelizing server-client computation (Zhang et al., 2023b). In addition, asynchronous architectures have been explored in other paradigms such as vertical (Zhang et al., 2024b) and clustered (Liu et al., 2024b) federated learning. While these directions complement our work, they fall outside the scope of this study.

## 3 PRELIMINARIES

### 3.1 ASYNCHRONOUS SYSTEM ARCHITECTURE

In an asynchronous federated learning system, a central server coordinates the training of a global model $W$ using data distributed across $M$ clients. Each client $m \in \{1, 2, \ldots, M\}$ possesses its own local dataset $\mathbf{D}_m$. The data distribution of client $m$ is denoted as $\mathcal{P}_m$. The objective is to train a global model $W$ that generalizes well across the combined data distribution of all clients. Formally, we aim to solve the following optimization problem:

$$W^* = \arg \min_{W} \frac{1}{M} \sum_{m=1}^{M} \mathbb{E}_{(x,y) \sim \mathcal{P}_m} \ell\left(f(x; W), y\right), \tag{1}$$

where $W$ denotes the global weights, $\ell$ the loss function, and $f(x; W)$ the prediction of the model on data $x$ with model weights $W$.

Clients perform local training and communicate their updates to the server at different times. Let $T$ be the number of global rounds. For $t \in \{1, \ldots, T\}$, denote $m_t \in \{1, \ldots, M\}$ as the client that communicates with the server at the $t$-th round, and $\tau_t$ as the round when client $m_t$ last communicated with the server. We define the staleness of the client update as follows:

**Definition 1** (Staleness). *Staleness quantifies the delay between a client's updates, defined as the number of global rounds between the client's last communication with the server and its current update. Formally, let $t$ be the current global round, and $\tau_t$ the global round when the server last received updates from client $m_t$. The staleness of client $m_t$ is defined as $t - \tau_t$, where $t - \tau_t \geq 1$. A staleness of 1 indicates no delay.*

For simplicity, we will drop the subscripts on $m_t$ and $\tau_t$ with no ambiguity from now on.

### 3.2 A MOTIVATING STUDY

We conduct an experiment on MNIST (Deng, 2012) with a LeNet5 (LeCun et al., 1998) model to analyze the challenges in asynchronous federated learning. We simulate the scenario with two clients: one with a 10-second latency and the other with 30, 60, or 100 seconds. We adopt the asynchronous method, FedAsync (Xie et al., 2019), where client updates are aggregated with decay factors based on latency. Let $W^{(t)}$ denote the global weights at $t$-th round before aggregation, and $W^{(t+)}$ the global weights after aggregation. Similarly, let $W_m^{(t)}$ represent the model weights of client $m$ at $t$-th round. The aggregation follows:

$$W^{(t+)} = (1 - \beta_t)W^{(t)} + \beta_t W_m^{(t)} \tag{2}$$

where $\beta_t = (t - \tau)^{-a} \cdot \beta$ with $\beta = 0.6$ and $a = 0.5$ following the setting in FedAsync. To create non-IID data, each client is assigned a non-overlapping half of the MNIST classes.

The global performance is shown in Figure 1(a), where markers represent updates from the "slow" client with longer latency. We observe an increase in accuracy when the server aggregates updates from the slow client, as these updates introduce knowledge of previously undertrained classes. However, this gain is gradually lost, with accuracy declining to around 0.5 after several updates from the faster client. This suggests the fast client's updates override the contributions of the slower client.

Moreover, as the latency of the slower client increases, the decay factor $\beta_t$ for integrating its updates decreases. This weakens its contribution to the global model and slows convergence, especially under non-IID data, as valuable knowledge from the slower client is not fully utilized.

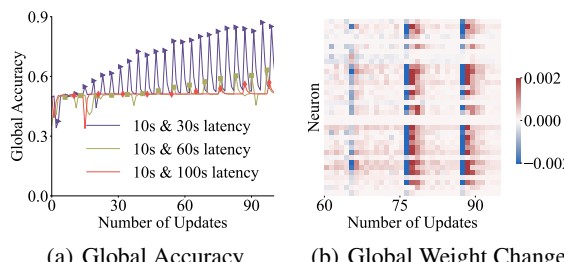

(a) Global Accuracy     (b) Global Weight Change

Figure 1(b) visualizes changes in global model weights in the last hidden layer in the case where the latency of the two clients is 10 and 100 seconds respectively. The y-axis represents neurons, and the x-axis represents the number of updates. The color indicates the direction and degree of global weight changes, with red representing an increase and blue a decrease. We observe abrupt shifts occur when switching between clients. Updates from the slow client often decrease the neuron weights (blue), while subsequent updates from the fast client increase the values (red), pulling the model in opposite directions. The antagonistic behavior is due to objective inconsistency—while the global model optimizes for the overall distribution, client training follow distinct local objectives, driving oscillations in weight aggregation.

Figure 1: Asynchronous learning with a fast client (10s latency) and a slow client (30/60/100s) assigned non-overlapping classes. Due to objective inconsistency: (a) accuracy spikes when the slow client updates, followed by drops as the fast client updates; (b) update directions shift abruptly when the active client switches.

# 4   METHOD: ORTHOGONAL WEIGHT CALIBRATION

## 4.1   ORTHOFL ALGORITHM

ORTHOFL maintains separate parameter sets for global and client models. Before merging global weight shift to the client model, it orthogonalizes the global shift against the client update. This orthogonality allows the client to absorb global progress in a way that reduces interference with its local optimization direction. Orthogonalization can be implemented at the server side, the client side, or through a hybrid approach, depending on practical deployment conditions. In our experiments, we implement orthogonalization on the server. The detailed discussion and pseudo-code are presented in Appendix B. Here, we describe the core algorithmic procedure.

**Global Aggregation via Moving Average**. Denote $W^{(t)}$ as the global model weights at the $t$-th round before client update and $W^{(t+)}$ after update. Similarly, let $W_m^{(t)}$ be the client $m_t$'s local model weights at the $t$-th round before update and $W_m^{(t+)}$ after update. Note that $W^{(t+1)} := W^{(t+)}$ as the global model weights stay unchanged after communication with a client before the next client update. We update the global model with a moving average:

$$W^{(t+)} = (1 - \beta_t)W^{(t)} + \beta_t W_m^{(t)}, \tag{3}$$

where $\beta_t$ controls the contribution of client $m$'s current update to the aggregation. We let $\beta_t := s_a(t - \tau) \cdot \beta$ with $\beta \in (0, 1)$ and $s_a(x) = x^{-a}$ for some $a > 0$ so that update with a larger staleness has a smaller contribution, and thereby decreasing the influence of client update with long delay.

**Calibration on Client Updates**. To reduce interference caused by asynchronous updates, ORTHOFL calibrates the local weights before the client starts the next round of local updates. The local weight change from its last update to its current update is calculated as:

$$\Delta_m = W_m^{(t)} - W_m^{(\tau+)}. \tag{4}$$

Similarly, the global weight shift during this period is calculated as:

$$\Delta = W^{(t)} - W^{(\tau+)}. \tag{5}$$

To integrate global progress into client $m$, ORTHOFL computes the orthogonal component of the global shift $\Delta$ with respect to client update $\Delta_m$ by removing the projection of $\Delta$ onto $\Delta_m$ in each

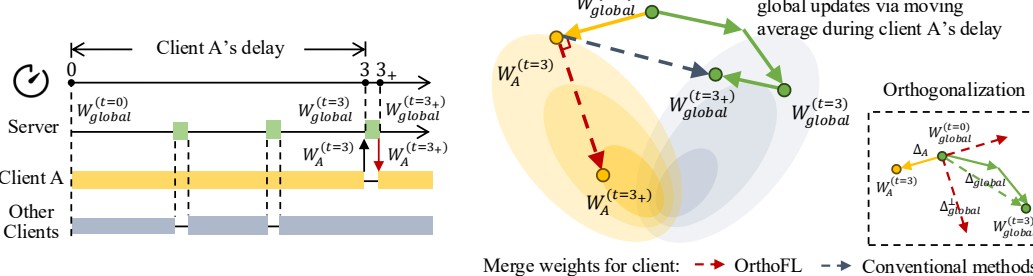

Figure 2: An example of optimization trajectories. Shaded regions show iso-loss contours for client A (yellow) and other clients (gray). Conventional methods that directly assign the aggregated global model (gray dashed line) can reverse client A's progress, pushing it away from lower-loss regions. ORTHOFL mitigates this interference by merging updates orthogonally (red dashed line).

layer. This projected component is excluded because it provides no new information (if aligned) or is contradictory to the local optimization (if opposed). This prevents redundant or disruptive weight changes and ensures the integrated knowledge is complementary to local learning.

While it is possible to perform orthogonalization at the full-model level by flattening all layer weights into a single vector, we adopt layer-wise projection for both structural and practical reasons. Structurally, it approximates a block-diagonal projection that respects the distinct roles of individual layers (e.g., extracting low-level features or high-level semantics), preventing the mixing of incompatible subspaces and preserving layer semantics. Practically, it is more memory-efficient, especially for large models, and supports modular, parallel computation by avoiding projections in extremely high-dimensional spaces. Formally, let $\Delta^l$ and $\Delta^l_m$ be the change in the layer $l$ of the global weights and the local weights respectively. The component of $\Delta^l$ orthogonal to $\Delta^l_m$ is:

$$\Delta^{l\perp} = \Delta^l - \text{proj}_{\Delta^l_m}(\Delta^l) = \Delta^l - \frac{\Delta^l \cdot \Delta^l_m}{\Delta^l_m \cdot \Delta^l_m}\Delta^l_m. \tag{6}$$

Let $\Delta^\perp$ represent the aggregation of $\Delta^{l\perp}$ across all layers. This orthogonal component $\Delta^\perp$ ensures that the updates from the other clients during delay $t - \tau$ do not interfere with the client's local progress, as it removes any component of the global weight change during delay along the direction of the local update. $\Delta^\perp$ is then added to the client model weight for the next round of local training:

$$W_m^{(t+)} = W_m^{(t)} + \Delta^\perp. \tag{7}$$

We present mathematical insights in Appendix C, which shows that our orthogonalization preserves the maximal component of the global shift within the hyperplane orthogonal to the client update.

## 4.2 VISUALIZATION OF OPTIMIZATION TRAJECTORIES

We illustrate the advantage of our method by visualizing an example of optimization trajectories. As shown in Figure 2, client A begins local training from the global state $W_{\text{global}}^{(t=0)}$. Before A's update arrives, the server aggregates two updates from other clients into the global model. Consequently, just before aggregating A's update at $t = 3$, the global model has evolved to $W_{\text{global}}^{(t=3)}$. Meanwhile, client A finishes local training and submits the updated parameters $W_A^{(t=3)}$. The shaded regions show the iso-loss contours for client A (yellow) and the collective optimization space of other clients (gray).

The global weight is updated via a moving average and becomes $W_{\text{global}}^{(t=3_+)}$. In conventional asynchronous methods, this global weight is directly assigned to client A (gray dashed line). This would push the model farther from A's optimization objective than $W_A^{(t=3)}$, reversing A's learning progress. ORTHOFL mitigates this by removing the component of the global weight shift that is parallel to $\Delta_A$. This ensures that the calibrated parameters are orthogonal to A's update direction. Finally, the calibrated global shift is merged into A's model (red dashed line), which becomes $W_A^{(t=3_+)}$. This

way, ORTHOFL reduces interference due to staleness and objective inconsistencies while preserving meaningful contributions at both global and local levels.

## 5 EXPERIMENTS

### 5.1 EXPERIMENT SETUP

**Compared Methods**. We consider seven baselines including synchronous methods, FedAvg (McMahan et al., 2017), FedProx (Li et al., 2020), FedAdam (Reddi et al., 2020), and asynchronous methods, FedAsync (Xie et al., 2019), FedBuff (Nguyen et al., 2022), CA$^2$FL (Wang et al., 2024a), and FADAS (Wang et al., 2024b). The details are in Appendix D.2.

**Applications and Datasets**. Table 1 summarizes the setups for the datasets. We conduct experiments on five datasets, including three data types: image, text, and time series. We pair each dataset with a model suited to its data type. To evaluate ORTHOFL's performance in parameter-efficient fine-tuning (PEFT) settings (Han et al., 2024), we employ a pretrained DistilBERT (Sanh, 2019) for evaluations on the 20 Newsgroups dataset and fine-tune it via Low-Rank Adaptation (LoRA) (Hu et al., 2021). Details are presented in Appendix D.1.

Table 1: Datasets and models in the experiments.

| Datasets | Clients | Avg. $|\mathbf{D}_m|$ | Model | Data Type |
|---|---|---|---|---|
| CIFAR-10 | 10 | 4000 | VGG11 | image |
| MNIST | 10 | 6000 | LeNet5 | image |
| 20 Newsgroups | 20 | 566 | DistilBERT | text |
| HAR | 21 | 350 | ResNet18 | time-series |
| CIFAR-100 | 100 | 400 | MobileNetV2 | image |

**Data Heterogeneity**. For the HAR dataset, clients are naturally divided based on the individual subjects, as each subject represents a client. For the other datasets, we set the number of clients equal to the number of classes in each dataset. To create non-IID client distributions, we follow prior work (Hsu et al., 2019) and use a Dirichlet distribution $Dir(\alpha = 0.1)$ to derive class distribution.

**Delay Simulation**. To ensure controlled evaluation, we simulate delays using measurements from prior work (Yu et al., 2023), collected on Raspberry Pi (RPi) devices in real home environments. We measure the round-wise training time for each dataset-model pair in our configuration on an RPi 4B, which has comparable hardware, and scale the computational and communication latencies relative to those in (Yu et al., 2023). From these scaled results, we derive the mean and variance of latency per device. Clients are randomly assigned latency profiles and sample delays at each round from a Gaussian distribution. All methods share the same client update order using a fixed set of random seeds for fair comparisons. Besides these real-world measurements, we also investigate model performance under additional delay distributions in Section 5.4.

**Evaluation Metrics**. We report accuracy after training for sufficient clock cycles to ensure the method reaches stable performance. For a fair comparison, we fix the same training time across all methods. In addition, we compare the time spent in reaching a target accuracy—set as the 95% of the lowest final accuracy among all compared methods. FedAvg serves as the baseline of time consumption (i.e., 1×), and we report the relative time for other methods.

### 5.2 MAIN EXPERIMENT RESULTS

As summarized in Table 2, ORTHOFL consistently outperforms the compared methods across all datasets—it not only converges faster but also achieves higher accuracy. Notably, the advantage becomes more obvious in the setting with a larger number of clients, as seen in CIFAR-100. This is because, for synchronous methods, the client sampling rate decreases as the number of clients increases, leading to longer wait times and slower convergence. Similarly, for baseline asynchronous methods, although their aggregation mechanisms are designed to mitigate the influence of model staleness, they fail to effectively address the challenges posed by data heterogeneity. In contrast, the calibration mechanism in ORTHOFL alleviates the negative impact of both stale model updates and the model divergence caused by data heterogeneity, ensuring faster convergence and improved performance. We conduct one-sided Wilcoxon-signed rank tests with Holm's $\alpha$ (5%) following previous works (Holm, 1979; Zhang et al., 2023a) and find that ORTHOFL significantly outperforms all baselines (the largest p-value is 0.007 after correction, far below 0.05). The accuracy curves over

Table 2: Main results (%) including average accuracy, standard deviation, and time relative to FedAvg. ORTHOFL reaches the target accuracy more quickly and achieves higher accuracy.

| Method | MNIST | | CIFAR-10 | | 20 Newsgroups | | HAR | | CIFAR-100 | |
|---|---|---|---|---|---|---|---|---|---|---|
| | Accuracy | Time | Accuracy | Time | Accuracy | Time | Accuracy | Time | Accuracy | Time |
| FedAvg | 92.2±1.1 | 1× | 73.8±2.4 | 1× | 58.1±3.1 | 1× | 84.2±0.6 | 1× | 22.2±0.4 | 1× |
| FedProx | 90.5±0.9 | 1.09× | 73.3±2.2 | 0.90× | 58.6±3.1 | 0.98× | 84.0±0.9 | 0.98× | 20.4±0.6 | 1.18× |
| FedAdam | 93.8±2.3 | 0.91× | 74.6±1.5 | 0.78× | 58.9±1.8 | 1.08× | 87.1±3.3 | 0.68× | 42.1±1.0 | 0.36× |
| FedAsync | 95.4±0.7 | 0.39× | 74.3±1.1 | 0.48× | 61.8±3.8 | 0.27× | 87.6±1.9 | 0.26× | 47.9±0.9 | 0.20× |
| FedBuff | 93.6±2.1 | 0.46× | 73.6±3.2 | 0.55× | 62.1±2.0 | 0.34× | 87.4±1.0 | 0.30× | 62.3±0.5 | 0.12× |
| CA²FL | 96.1±1.4 | 0.30× | 69.7±8.0 | 0.60× | 65.7±1.6 | 0.24× | 87.8±1.9 | 0.29× | 61.1±0.7 | 0.09× |
| FADAS | 93.9±0.9 | 0.98× | 71.6±1.3 | 1.01× | 63.5±2.3 | 0.49× | 88.3±1.7 | 0.50× | 49.5±0.4 | 0.22× |
| OrthoFL | **98.2±0.3** | **0.18×** | **76.5±3.3** | **0.19×** | **66.2±1.3** | **0.09×** | **89.7±1.0** | **0.23×** | **63.0±0.6** | **0.03×** |

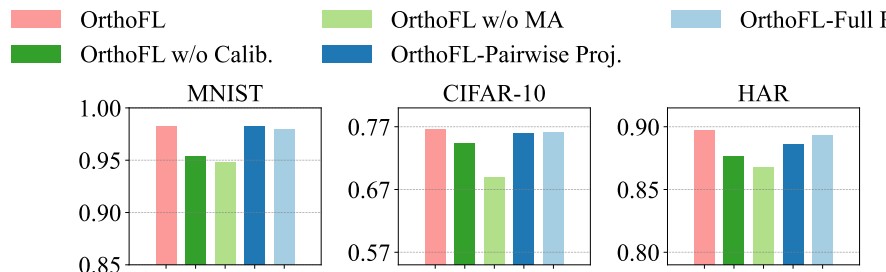

Figure 3: Ablation studies: Removing calibration (w/o Calib.) or global moving average (w/o MA) reduces performance. Pairwise projection variant (Pairwise Proj.) shows competitive performance, suggesting that effective orthogonal calibration can be achieved via multiple viable approaches. Layer-wise projection in ORTHOFL outperforms the full-model projection variant (Full Proj.).

training time are presented in Appendix E and hyperparameter sensitivity analyses are in Appendix F. ORTHOFL is robust to the variation of hyperparameters.

## 5.3 ABLATION STUDIES

We conduct ablation studies to evaluate our key design choices. (1) We set FedAsync as a baseline (ORTHOFL w/o Calib.), since it is equivalent to removing calibration. (2) We assess the contribution of global aggregation by removing the moving average and directly loading the calibrated client weight into the global model (ORTHOFL w/o MA). (3) We investigate an alternative orthogonalization strategy that projects the incoming client update onto the orthogonal subspace of the most recent updates from all other clients (ORTHOFL-Pairwise Proj.). (4) We replace the layer-wise projection in ORTHOFL with full-model projection which flattens all parameters into a single vector (ORTHOFL-Full Proj.) The orthogonality is achieved through the same process as in ORTHOFL.

Performance drops when clients and the global model share the same weights (ORTHOFL w/o Calib. and ORTHOFL w/o MA). ORTHOFL w/o MA performs particularly poorly as the absence of a global moving average causes overfitting to local distributions. These results confirm the importance of decoupling global and local learning. ORTHOFL-pairwise Proj. achieves results comparable to ORTHOFL, suggesting that orthogonal calibration can be effective in multiple ways. We expect that optimizing the orthogonalization strategy could bring further improvement, and we leave such exploration for future work. ORTHOFL's layer-wise projection performs better than full-model projection, being both more memory-efficient and better at preserving the semantic structure of individual layers.

## 5.4 EXPLORATORY STUDIES

**How do algorithms perform under different delay distributions?** Since real-world deployment of federated learning may present diverse delay patterns, we explore other possible delay distributions in real-world setups, such as following log-normal, half-normal (Sui et al., 2016), and uniform

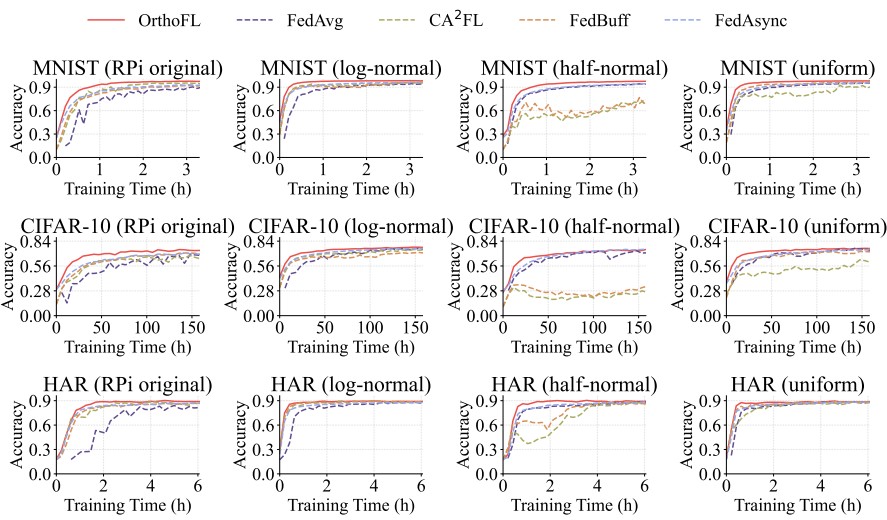

Figure 4: Performance with different delay distributions.

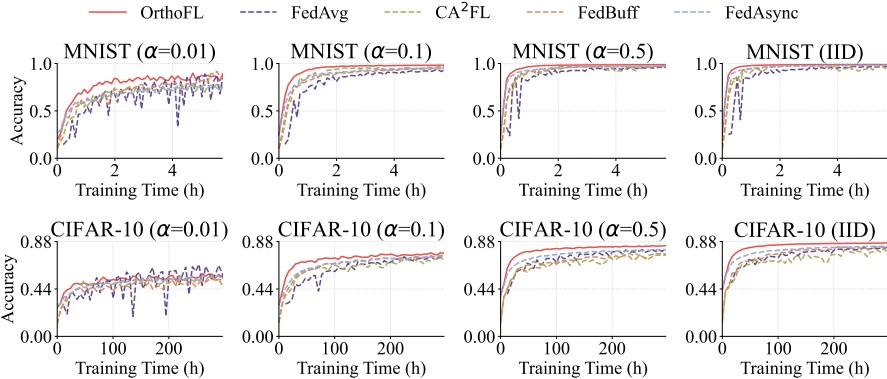

Figure 5: Performance under different data heterogeneity levels.

distributions (Nguyen et al., 2022). Each distribution is parameterized by the mean and variance of latency (communication and computation) observed across the RPi devices. Details on deriving these distributions are in Appendix D.4.

Figure 4 presents the accuracy curves for each algorithm under different delay distributions. With real-world measured RPi latency, some clients experience substantially longer latencies, causing synchronous methods like FedAvg to converge more slowly as the server waits for stragglers. In this case, asynchronous methods generally have better performance than synchronous ones. Under the lognormal, half-normal, and uniform delay distributions, extreme latencies are less common, so the performance gap between synchronous and asynchronous methods narrows. In general, ORTHOFL performs the best across all scenarios, demonstrating its robustness against different delay patterns.

**How does data heterogeneity impact performance?** To control the level of data heterogeneity, we change $\alpha$ for Dirichlet distribution from $\{0.01, 0.1, 0.5, 10^4\}$, where $\alpha = 10^4$ simulates the IID case. We present experiments on MNIST and CIFAR-10 as shown in Figure 5. As the client data distribution becomes more heterogeneous (i.e., lower values of $\alpha$), we observe the performance of baseline methods has more fluctuations and decreases in final accuracy. This is because client models trained on non-IID data distributions have larger divergences in their weights. Aggregating divergent updates amplifies inconsistencies, leading to slower convergence and lower accuracy for baseline methods. By contrast, ORTHOFL exhibit stable performance across all settings. In the IID case, where data is uniformly distributed across clients, ORTHOFL still outperforms the compared methods. This is because our orthogonal calibration also addresses model staleness.

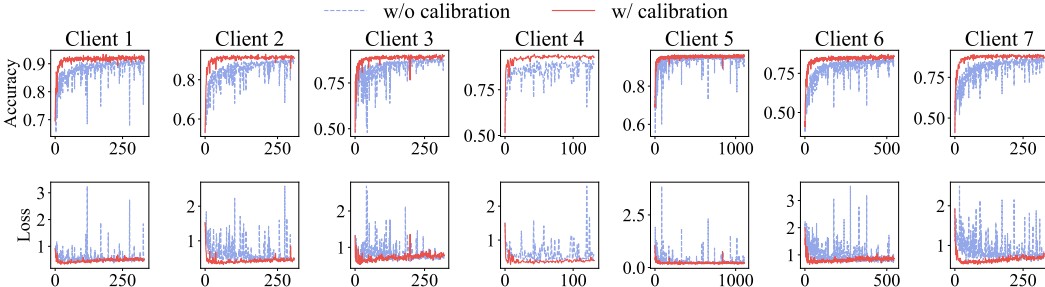

Figure 6: Client-wise accuracy and loss (y-axis) on local validation data over the number of updates received (x-axis). ORTHOFL improves local stability by reducing interference from global updates.

## 5.5 CASE STUDY: INTERFERENCE IN LOCAL TRAINING

To evaluate how orthogonal calibration mitigates interference, we present a case study on CIFAR-10 showing accuracy and loss trajectories for a subset of clients (others exhibit similar patterns). Specifically, we track each client's model performance on a held-out local validation set over the number of global updates it receives. This provides a direct measure of how merging the global weights affects local optimization.

As shown in Figure 6, without calibration, clients frequently experience spikes in loss and sharp drops in accuracy—evidence of interference from conflicting global updates. In contrast, ORTHOFL stabilizes local training across all clients as accuracy curves remain high and smooth, and loss curves stay low with minimal perturbation. This demonstrates that orthogonal calibration effectively reduces harmful interference from global updates and better preserves local learning progress.

## 5.6 CALIBRATION OVERHEAD ANALYSIS

The calibration is lightweight because it only requires basic vector operations (dot products and subtractions), in contrast to the repeated forward and backward passes of model training. Hence, it is efficient even on low-capacity devices. Its time complexity is $O(P)$, where $P$ is the number of model parameters. It is independent of the number of clients, since the projection is computed once per client update. For comparison, conventional asynchronous aggregation also has $O(P)$ complexity, while client training (i.e., gradient descent) typically requires $O(EBP)$ opera-

Table 3: Calibration time of ORTHOFL.

| Dataset | Model | Train Params | Time |
|---|---|---|---|
| MNIST | LeNet5 | 44K | 0.003s |
| HAR | ResNet18 | 119K | 0.199s |
| 20 Newsgroups | DistilBERT | 753K | 0.110s |
| CIFAR-10 | VGG11 | 9.2M | 0.188s |
| CIFAR-100 | MobileNetV2 | 2.4M | 0.512s |

tions per round, where $E$ is the number of local epochs and $B$ is the number of mini-batches. Table 3 reports the calibration time including orthogonalization and updating the global weight state, when running on a node equipped with an AMD EPYC 7713 64-Core Processor (3.72 GHz max clock) and 3.9 TiB RAM. This cost is negligible compared to the operational latencies on clients (in Figure 7). Efficiency can be further improved by layer-wise parallelism and approximate projection.

## 6 CONCLUSIONS

In this paper, we introduce ORTHOFL, an orthogonal calibration mechanism for asynchronous federated learning. ORTHOFL decouples the global and local learning progress and employs a geometric projection to calibrate and integrate global progress into client models without disrupting local learning. Experiments demonstrate that ORTHOFL consistently outperforms state-of-the-art synchronous and asynchronous baselines, achieving notable gains in accuracy, convergence speed, and robustness under diverse delay patterns and data heterogeneity. For future work, we plan to investigate the long-term training dynamics of orthogonal calibration, including its effects on stability, forgetting, and distribution shift. We also aim to further optimize learning efficiency, for example, through integrating adaptive client selection into ORTHOFL to prioritize impactful clients for participation.

## ETHICS STATEMENT

This work contributes foundational advances in federated learning and is not tied to specific applications, thus raising no ethical concerns.

## REPRODUCIBILITY STATEMENT

All datasets used in this paper are publicly available, with references provided in Appendix D.1. Source code is included in the supplemental material. Detailed experimental settings, including hyperparameters, federated system configurations, and delay simulation, are described in Section 5.1 and Appendices D.3, D.4. The hardware and compute resource details are specified in Section 5.1 and 5.6.

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

## A  THE USE OF LARGE LANGUAGE MODELS (LLMs)

The use of LLMs in this work is limited to polishing text and formatting. They do not contribute to the ideation or methodological development.

## B  PSEUDO-CODE OF ORTHOFL

In practice, ORTHOFL can be implemented with orthogonalization performed on the server side, the client side, or through a hybrid approach. The choice depends on the availability of server storage and the computational capability of participating nodes. When orthogonalization is handled by the server, the server maintains a copy of the global state $W^{(\tau+)}$ for a client $m$ corresponding to when it sent the model to the client. Once the client returns its update, the server retrieves the stored $W^{(\tau+)}$, computes the global shift, and applies orthogonal calibration. Alternatively, $W^{(\tau+)}$ can be stored on the client and communicated back to the server to reduce server-side storage requirement. The pseudo-code of ORTHOFL under the server orthogonalization protocol is presented in Algorithm 1.

---

**Algorithm 1:** ORTHOFL Framework (Server-Side Orthogonalization)

---

**Input**  : Total number of updates $T$, local training epochs $E$, initial global model weights $W^{(0)}$.
**Output** : Global model weights $W^{(T+)}$.
**Server execution:**
Send $W^{(0)}$ to all available clients;
**for** $t = 1, \ldots, T$ **do**
    Receive update $W_m^{(t)}$ from a client;
    Compute global weight shift: $\Delta = W^{(t)} - W^{(\tau+)}$ ;
    Compute client weight shift: $\Delta_m = W_m^{(t)} - W_m^{(\tau+)}$ ;
    Update global model: $W^{(t+)} = (1 - \beta_t)W^{(t)} + \beta_t W_m^{(t)}$;
    Orthogonalize each layer: $\Delta^{l\perp} = \Delta^l - \frac{\Delta^l \cdot \Delta_m^l}{\Delta_m^l \cdot \Delta_m^l}\Delta_m^l$;
    Merge weights for client: $W_m^{(t+)} = W_m^{(t)} + \Delta^\perp$;
    ClientUpdate$(m, W_m^{(t+)})$
**return** $W^{(T+)}$;
**ClientUpdate**$(m, \tilde{W})$:
$W_m \leftarrow \tilde{W}$;
**for** $e = 1, \ldots, E$ **do**
    Partition $\mathbf{D}_m$ into mini-batches $\{B_i^{(m)}\}_{i=1}^{j_m}$;
    **for** $i = 1, \ldots, j_m$ **do**
        Update local weights: $W_m \leftarrow W_m - \eta_c \nabla_{W_m} \mathcal{L}_m(W_m; B_i^{(m)})$;
Compute update: $\Delta_m = W_m - \tilde{W}$;
**return** $W_m$ to server;

---

As discussed in Section 5.6, the cost of orthogonalization is marginal compared to local training, which makes it practical to delegate this step to clients. Algorithm 2 presents the pseudo-code of ORTHOFL under the local orthogonalization protocol. Each client derives its stored global state $W^{(t+)}$ by applying the aggregation rule to the received global weight $W^{(t)}$. Before local training in the next round, it computes the global shift (i.e., the difference between the global model at the start and end of the previous round) and applies orthogonal calibration to adjust its initialization. This design avoids storing global state history on the server, making it more scalable in terms of disk space.

---

**Algorithm 2:** ORTHOFL Framework (Client-Side Orthogonalization)

---

**Input** : Total number of updates $T$, local training epochs $E$, initial global model weights $W^{(0)}$.

**Output** : Global model weights $W^{(T+)}$.

**Server execution:**

Send $W^{(0)}$ to all available clients;

**for** $t = 1, \ldots, T$ **do**

    Receive updated $W_m^{(t)}$ from a client;

    ClientUpdate($m, \beta_t, W^{(t)}$);

    Update global model: $W^{(t+)} = (1 - \beta_t)W^{(t)} + \beta_t W_m^{(t)}$;

**return** $W^{(T+)}$;

**ClientUpdate**($m, \beta_t, \tilde{W}_{\text{global}}$):

`// Input` $\tilde{W}_{\text{global}} = W^{(t)}$ `(global state when last training finishes)`

`// Stored` $W_{\text{global}} = W^{(\tau+)}$ `(global state when last training starts)`

Compute global shift: $\Delta = \tilde{W}_{\text{global}} - W_{\text{global}}$;

Update global state: $W_{\text{global}} = (1 - \beta_t)\tilde{W}_{\text{global}} + \beta_t W_m$ ;     `//` $W_{\text{global}}$ `becomes` $W^{(t+)}$

Orthogonalize each layer: $\Delta^{l\perp} = \Delta^l - \frac{\Delta^l \cdot \Delta_m^l}{\Delta_m^l \cdot \Delta_m^l} \Delta_m^l$;

Merge weights for client: $W_m^{\text{init}} \leftarrow W_m + \Delta^\perp$;

Initialize local weights: $W_m \leftarrow W_m^{\text{init}}$;

**for** $e = 1, \ldots, E$ **do**

    Partition $\mathbf{D}_m$ into mini-batches $\{B_i^{(m)}\}_{i=1}^{j_m}$;

    **for** $i = 1, \ldots, j_m$ **do**

        Update local weights: $W_m \leftarrow W_m - \eta_c \nabla_{W_m} \mathcal{L}_m(W_m; B_i^{(m)})$;

Compute client weight change: $\Delta_m = W_m - W_m^{\text{init}}$ ;

**return** $W_m$ to server;

---

Moreover, the two protocols can be combined. For example, clients with limited resources may delegate orthogonalization to the server, while more capable clients perform it locally. Such a hybrid approach allows the system to accommodate heterogeneous device capacities.

## C  MATHEMATICAL BASIS OF ORTHOGONALIZATION

A gradient update orthogonal to previously accumulated gradients helps preserve existing model behavior and minimizes unintended changes to its outputs (Farajtabar et al., 2020). Due to the high-dimensional parameter space of neural networks, there are multiple directions that are orthogonal to the stale local weight update. We have the following lemma showing that the orthogonalization strategy in ORTHOFL preserves the maximal information from the global weight shift vectors perpendicular to the local update direction.

For $v, w \in \mathbb{R}^d$, let $\langle v, w \rangle := \sum_{i=1}^d v_i w_i$ be the standard inner product on $\mathbb{R}^d$.

**Lemma C.1.** *Let $v \in \mathbb{R}^d$ and $\mathcal{U} = \{u_1, \cdots, u_k\}$ be an orthogonal set for some $k < d$. Then for any $w \in (\text{span}\,\mathcal{U})^\perp$,*

$$\|v - v^\perp\| \leq \|v - w\|, \tag{8}$$

*where $v^\perp := v - \sum_{i=1}^k \langle v, u_i \rangle \frac{u_i}{\|u_i\|^2}$ denote the component of $v$ orthogonal to $\mathcal{U}$. Moreover, the angle between $v$ and $v^\perp$ is less than any angle between $v$ and $w$ for $w \in (\text{span}\,\mathcal{U})^\perp$.*

We apply this lemma in our case where $v = \Delta^l$ and $\mathcal{U} = \{\Delta_m^l\}$. It implies $\Delta^{l\perp}$ in equation (6) is the unique vector among those perpendicular to $\Delta_m^l$ with the smallest magnitude of $\|\Delta^l - \Delta^{l\perp}\|$ and the smallest angle with $\Delta^l$. This allows the server to transfer the maximum amount of global progress to client $m$, among all projections onto the orthogonal subspaces of the client update.

*Proof.* By normalizing the vectors in $\mathcal{U}$, we may assume $\mathcal{U}$ is orthonormal. Extend $\mathcal{U}$ to an orthonormal basis $\mathcal{B} := \{u_1, \cdots, u_k, u_{k+1}, \cdots, u_d\}$ on $\mathbb{R}^d$. Write $w = \sum_{i=1}^{d} w_i u_i$ and $v = \sum_{i=1}^{d} v_i u_i$, where $w_i := \langle w, u_i \rangle$ and $v_i := \langle v, u_i \rangle$ denote the coordinates of $w$ and $v$ with respect to the basis $\mathcal{B}$ respectively. Since $w \in (\mathrm{span}\,\mathcal{U})^\perp$, $w_i = 0$ for $1 \leq i \leq k$. It follows from Pythagorean theorem that

$$
\begin{aligned}
\|v - w\|^2 &= \left\| \sum_{i=1}^{d} (v_i - w_i) u_i \right\|^2 \\
&= \left\| \sum_{i=1}^{k} v_i u_i + \sum_{j=k+1}^{d} (v_j - w_j) u_j \right\|^2 \\
&= \left\| \sum_{i=1}^{k} v_i u_i \right\|^2 + \left\| \sum_{j=k+1}^{d} (v_j - w_j) u_j \right\|^2 \\
&\geq \left\| \sum_{i=1}^{k} v_i u_i \right\|^2 .
\end{aligned}
\tag{9}
$$

Taking square root of both sides and noting that by definition $v - v^\perp = \sum_{i=1}^{k} v_i u_i$, equation (8) follows. As for the second claim, let $\theta(v, w)$ denote the angle between $v$ and $w$, $\theta(v, v^\perp)$ the angle between $v$ and $v^\perp$. By the Cauchy-Schwarz inequality,

$$
\begin{aligned}
\langle v, w \rangle &= \left\langle v, \sum_{j=k+1}^{d} w_j u_j \right\rangle \\
&= \sum_{j=k+1}^{d} w_j \langle v, u_j \rangle \\
&\leq \left( \sum_{j=k+1}^{d} w_j^2 \right)^{1/2} \cdot \left( \sum_{j=k+1}^{d} v_j^2 \right)^{1/2} \\
&= \|w\| \cdot \|v^\perp\|.
\end{aligned}
\tag{10}
$$

It follows that

$$
\cos \theta(v, w) := \frac{\langle v, w \rangle}{\|v\|\,\|w\|} \leq \frac{\|v^\perp\|}{\|v\|} = \frac{\langle v, v^\perp \rangle}{\|v\|\,\|v^\perp\|} = \cos \theta(v, v^\perp).
\tag{11}
$$

Since the cosine function is monotonically decreasing on $[0, \pi]$, $\theta(v, w) \geq \theta(v, v^\perp)$ as claimed. □

## D    DETAILS OF EXPERIMENT SETUP

### D.1    APPLICATIONS AND DATASETS

The experiments are conducted with the following three applications.

1. **Image Classification.** We evaluate our framework on three widely used image datasets: MNIST (Deng, 2012), CIFAR-10, and CIFAR-100 (Krizhevsky et al., 2009). For MNIST, we use LeNet5 (LeCun et al., 1998), a lightweight convolutional network. For CIFAR-10, we adopt VGG11 (Simonyan and Zisserman, 2014), a deeper convolutional architecture. For CIFAR-100, we employ MobileNetV2 (Sandler et al., 2018), a compact and efficient model ideal for large-scale image classification tasks.

2. **Text Classification.** We experiment with the 20 Newsgroups dataset (Lang, 1995), a benchmark dataset for multi-class text categorization. We adopt DistillBERT (Sanh, 2019), a small transformer model suitable for resource-constrained devices. The pretrained weights are from Hugging Face[1].

---

[1] https://huggingface.co/distilbert/distilbert-base-uncased

We use the HAR (Anguita et al., 2013) dataset, which contains time-series sensor data for different physical activities. We adopt the 1D version of ResNet18 (He et al., 2016), a modified ResNet architecture for processing 1D sequential data.

3. **Human Activity Recognition.** We use the HAR (Anguita et al., 2013) dataset, which contains time-series sensor data for different physical activities. We adopt the 1D version of ResNet18 (He et al., 2016), a modified ResNet architecture for processing 1D sequential data.

All datasets provide predefined training and test splits. For HAR, we use the test set for global evaluation and assign each subject's training data to an individual client. For the other datasets, we also use the test set for evaluation and sample client data from the training set using a Dirichlet distribution (Section 5.1).

## D.2 BASELINE METHODS

Below are the baseline methods that we compared in the experiments:

- **FedAvg** (McMahan et al., 2017) is the classical synchronous algorithm where the server selects a subset of clients to conduct training in each round and synchronizes updates from these clients before aggregation.

- **FedProx** (Li et al., 2020) is a synchronous method that addresses data heterogeneity by incorporating L2 regularization during local training to constrain the divergence between global and client models.

- **FedAdam** (Reddi et al., 2020) is a synchronous algorithm that integrates Adam optimizer for the server. It adapts learning rates for each parameter using first- and second-moment estimates, improving convergence and accelerating training under data heterogeneity.

- **FedAsync** (Xie et al., 2019) is a fully-asynchronous framework that lets the server immediately aggregate the client updates into the global model upon receipt. It uses a weighting mechanism as in Equation 2 to account for the staleness of the updates.

- **FedBuff** (Nguyen et al., 2022) is a semi-asynchronous framework that introduces a buffered aggregation strategy. It maintains a buffer to collect client updates. Once the buffer is full, the server aggregates the client updates in the buffer and updates the global model.

- **CA$^2$FL** (Wang et al., 2024a) is another semi-asynchronous framework based on buffered aggregation. It caches the latest update from every client on the server and uses them to estimate the clients' contribution to the update of the current round and calibrate global updates.

- **FADAS** (Wang et al., 2024b) extends adaptive federated optimization methods (e.g., FedAdam) to the asynchronous setting, and further introduces a delay-adaptive learning rate that adjusts the global step size based on the staleness of client updates.

## D.3 FEDERATED LEARNING CONFIGURATION

For reproducibility, we report the configurations in our experiments. The number of local training epochs $E = 5$. For the FedAvg algorithm, the number of sampled clients at each round is 10. The aggregation hyperparameters in Equation 3 are set to $\beta = 0.6$ and $a = 0.5$, following the values used in prior work (Xie et al., 2019). The learning rate for local training at all clients is $5 \times 10^{-5}$ for the 20 Newsgroups dataset and 0.01 for the other datasets.

## D.4 DERIVATION OF DELAY DISTRIBUTIONS

**Deployment Specificities in Delay Collection**. The prior work (Yu et al., 2023) collected computation and communication latency on RPis when the devices were training a convolutional neural network (2 convolutional layers) on the MNIST dataset in a federated learning setup over 100 rounds. To account for differences in the size of models and datasets, we use an RPi 4B which has comparable computational capabilities as the reported ones to measure latency for training one round under each model and dataset configuration. Each configuration is tested five times to derive the average training time. The computational latency for datasets in our experiment is then adjusted based on the ratio between our measured time and the computation latency in (Yu et al., 2023). Similarly,

communication latency was scaled based on the model size in bytes compared to the model in the original delay collection. Figure 7 presents the average communication and computation latency on Raspberry Pis scaled for the dataset and model configuration in our experiments. For 20 Newsgroups, the converted computational time is sufficiently long, making communication time negligible.

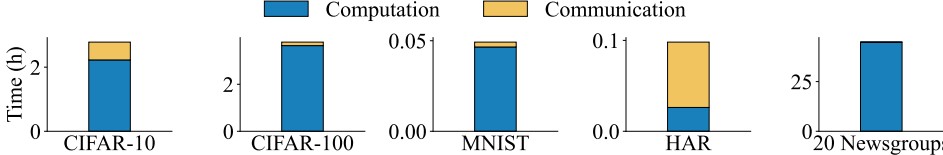

Figure 7: Average latency per round across clients.

**Additional Delay Distributions**. In Section 5.4, we present the performance of compared methods under additional delay distributions following physical deployments. The derivations of the additional delay distributions are as follows.

- **Lognormal distribution**: The parameters $\mu$ (mean) and $\sigma$ (standard deviation) of the natural logarithm of delays are derived from the measurements on Raspberry Pis. Specifically, $\mu_R$ and $\sigma_R$ represent the arithmetic mean and standard deviation of the measured delays for all rounds, respectively.

$$\sigma = \sqrt{\ln\left(\frac{\sigma_R^2}{\mu_R^2} + 1\right)}, \quad \mu = \ln(\mu_R) - \frac{\sigma^2}{2}$$

Then, the latency for each client is sampled from the lognormal distribution to capture skewed and heavy-tailed delays.

- **Half-normal distribution**: The mean and standard deviation of the delays (i.e., $\mu_R$ and $\sigma_R$) are calculated from the delay measurements on Raspberry Pis. Then, for each client, its latency is sampled from the half-normal distribution, ensuring non-negative values and a skewed distribution toward smaller delays.

- **Uniform distribution**: Client latency is sampled from a uniform distribution with bounds set between the 5th and 95th percentiles of the measurements from Raspberry Pis. This ensures outliers are excluded while covering the majority of the observed range.

Figure 8 shows the simulated latency for 100 clients, each running CIFAR-100 with MobileNetV2 over 10,000 rounds.

## E  CONVERGENCE STABILITY OVER TIME

Figure 9 shows the accuracy of each method over training time. Compared to baseline methods, ORTHOFL achieves faster convergence and significantly reduces accuracy fluctuations over time. The instability observed in other methods is primarily caused by conflicting updates due to data heterogeneity and asynchronous optimization, which ORTHOFL mitigates through orthogonal calibration.

## F  SENSITIVITY ANALYSES

**Aggregation Hyperparameter**. In Equation 3, the parameter $\beta$ scales the overall contribution of the client update to the global model, and $a$ controls the sensitivity to staleness. A larger $\beta$ allows the global model to incorporate more of the client updates, potentially accelerating learning, while a smaller $\beta$ preserves more of the global model's previous state, enhancing stability. The exponent $a$ modulates the decay of $\beta_t$ with respect to staleness: higher $a$ values cause staler updates to contribute less. We conduct a sensitivity analysis by varying $\beta \in \{0.2, 0.4, 0.6, 0.8\}$ and $a \in \{0.3, 0.5, 0.7\}$. As shown in Table 4, ORTHOFL is robust to different $a$ and $\beta$ values and always achieves higher accuracy than the baselines after a fixed training time, and the relative time of ORTHOFL is low.

**Number of Local Training Epochs**. The number of local training epochs $E$ determines how much information is learned during a round and impacts the overall convergence speed. A larger $E$ allows

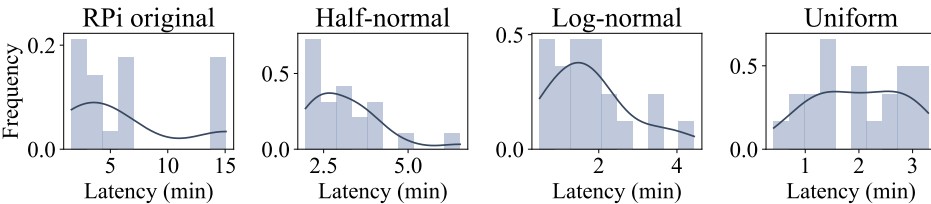

Figure 8: Delay patterns under different distributions: measurements from RPis show discrete peaks and high variability, half-normal and lognormal distributions have long tails, and uniform distribution assumes equal probability within a bounded range.

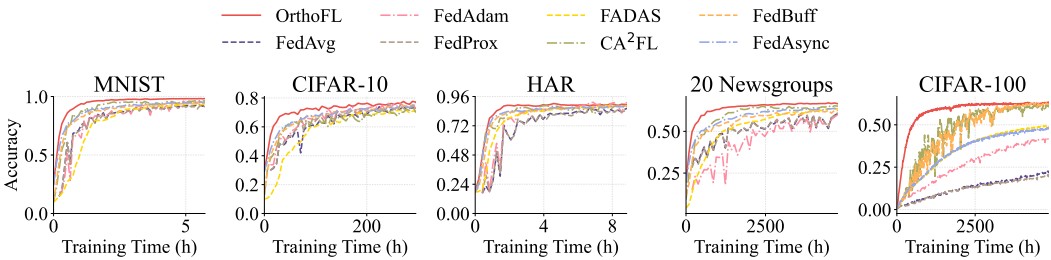

Figure 9: Accuracy over training time. ORTHOFL converges faster and exhibits more stable performance compared to baselines.

clients to learn more information from their local data, potentially improving local model performance. However, it has the risk of larger divergence among client models and global models. On the other hand, setting a smaller $E$ ensures closer alignment between client and global models but comes at the cost of higher communication latency due to more frequent synchronization. We vary the value of $E$ from $\{1, 5, 10\}$ and show the results in Figure 10. The upper row shows the final accuracy after a fixed training time and the bottom row presents the relative time to reach 95% of FedAvg's target accuracy when $E = 5$. We observe that when $E = 1$, both methods reach lower accuracy compared to larger $E$. This is due to insufficient local learning, requiring more communication rounds to achieve comparable performance. ORTHOFL achieves similar final accuracy when $E = 5$ and $E = 10$. Notably, ORTHOFL outperforms FedAvg across different $E$ values. The speedup of ORTHOFL is more obvious at smaller $E$ values (e.g., $E = 1$).

## G    CONNECTIONS WITH OTHER AREAS

**Asynchronous Stochastic Gradient Descent**. Asynchronous stochastic gradient descent (SGD) is closely related to asynchronous federated learning and has provided theoretical and empirical foundations for scalable distributed training. Early studies analyzed error-runtime trade-offs, showing that incorporating stale gradients can alleviate system bottlenecks without significantly compromising accuracy (Dutta et al., 2018). Subsequent work refined convergence bounds based on maximum (Stich and Karimireddy, 2019) or average (Koloskova et al., 2022) delay and demonstrated that asynchronous SGD can converge faster than traditional minibatch SGD (Mishchenko et al., 2022). To tackle challenges such as gradient staleness, communication delays, and convergence guarantees, various strategies have been proposed, such as filtering out outlier gradients (Xie et al., 2020; Cohen et al., 2021), adjusting update steps according to delay (Mishchenko et al., 2018; Aviv et al., 2021), and approximating gradients to compensate for delayed information (Zheng et al., 2017). However, unlike federated learning, most asynchronous SGD formulations do not explicitly address non-i.i.d. data distributions or the strict data privacy constraints inherent in federated settings, which limits their direct applicability.

**Orthogonal Gradient Descent**. There are works in continual learning (Farajtabar et al., 2020; Chaudhry et al., 2020; Saha et al., 2021; Lin et al., 2022; Chen et al., 2022) that leverage the idea of orthogonalization to project gradients onto non-conflicting subspaces across tasks. The goal is

Table 4: Sensitivity analysis on the aggregation hyperparameters $a$ and $\beta$. ORTHOFL demonstrates robustness to variations in the two hyperparameters, maintaining high accuracy and fast training speed (measured as time relative to FedAvg).

| Dataset | Metric | $\beta$ ($a = 0.5$) | | | | $a$ ($\beta = 0.6$) | | |
|---------|--------|------|------|------|------|------|------|------|
| | | 0.2 | 0.4 | 0.6 | 0.8 | 0.3 | 0.5 | 0.7 |
| MNIST | Accuracy | 98.3±0.2 | 98.3±0.3 | 98.2±0.3 | 98.0±0.4 | 98.0±0.6 | 98.2±0.3 | 98.2±0.3 |
| | Time | 0.19× | 0.18× | 0.18× | 0.19× | 0.20× | 0.18× | 0.19× |
| CIFAR10 | Accuracy | 77.8±2.6 | 77.3±3.2 | 76.5±3.3 | 77.9±1.9 | 76.2±4.2 | 76.5±3.3 | 76.8±3.5 |
| | Time | 0.17× | 0.17× | 0.19× | 0.23× | 0.20× | 0.19× | 0.19× |
| HAR | Accuracy | 88.5±2.3 | 89.3±1.7 | 89.7±1.0 | 88.3±0.8 | 88.2±1.1 | 89.7±1.0 | 89.8±1.4 |
| | Time | 0.20× | 0.19× | 0.23× | 0.18× | 0.19× | 0.23× | 0.18× |

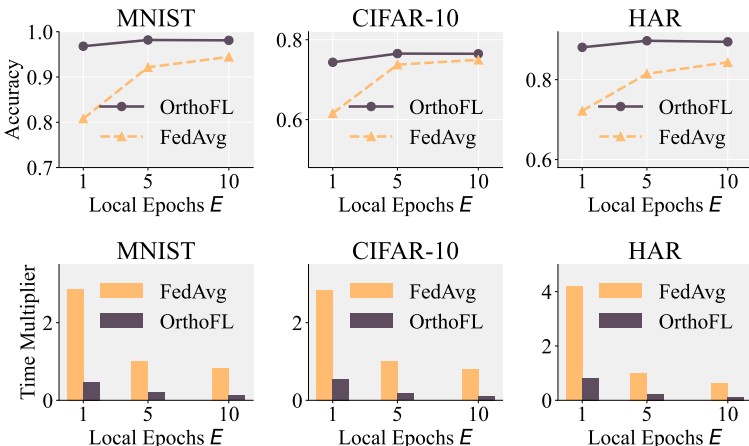

Figure 10: Sensitivity analysis on the number of local training epochs $E$. An appropriately chosen $E$ improves accuracy within the same training duration and expedites convergence.

to prevent catastrophic forgetting, where new knowledge overrides previously learned information when training a model on a sequence of tasks. It shares a common goal with asynchronous federated learning, which is to mitigate knowledge interference. The earliest formulation (Farajtabar et al., 2020) proposes projecting new gradients onto the space orthogonal to all previously stored gradient directions from old tasks. Chaudhry et al. (2020) extends this by assigning each task a low-rank orthogonal projection and trains the network via Stiefel manifold optimization to ensure orthonality. Saha et al. (2021) identifies important gradient subspaces by applying singular value decomposition (SVD) on hidden activations and constrains new gradients to be orthogonal to these subspaces. Lin et al. (2022) further enhances this approach by defining a trust region of relevant past tasks and selectively reusing their subspaces through scaled projections, enabling a balance between preserving old knowledge and transferring to new tasks. Despite the similarities, the two fields follow distinct training paradigms. Continual learning typically processes tasks sequentially in a centralized setting. Methods can reuse data or access sample-level information (e.g., gradient, representations) from previous tasks. In contrast, federated learning involves repeated rounds of training from distributed clients with data constraints. The model must integrate updates without knowing data or gradients from clients to preserve privacy. Therefore, the two fields require different optimization techniques and system designs.

## H  ADAPTABILITY TO DYNAMIC ENVIRONMENTS

The asynchrony and orthogonal calibration mechanisms not only accelerate training but also extend its applicability to more complex scenarios. These features make ORTHOFL suitable in dynamic federated learning environments. We discuss the following situations:

**Dynamic Client Participation**. In real-world federated learning applications, it is common for new clients to join the training process (Park et al., 2021) or for previously unseen data with new tasks to appear over time (Yang et al., 2024; Yoon et al., 2021; Ma et al., 2022; Dong et al., 2022). Scalability to new clients and tasks is essential for long-term performance. ORTHOFL is readily applicable to these settings. By orthogonal calibration on asynchronous updates, ORTHOFL mitigates the disruptive impact of newly joined clients, especially if the global model is well-trained and initial updates from newly joined clients are noisy or biased due to limited data or distribution shifts. Similarly, when new classes are introduced, ORTHOFL preserves knowledge of previously learned classes, reducing forgetting and performance degradation caused by sudden distribution shifts.

**Failure Handling**. Large-scale federated learning faces a high risk of client or system failures due to issues such as out-of-memory (OOM) errors during training, hardware malfunctions, or network interruptions (Jang et al., 2023; Gupta et al., 2017; Jeon et al., 2019; Weng et al., 2022), particularly when training large models across many devices. Failure rates tend to increase with larger models and more clients, as the computational and communication demands scale up significantly. The ability to effectively handle these failures is critical for ensuring robust and reliable training in federated learning systems. The asynchronous nature of ORTHOFL makes it robust to such failures. When a client fails, training continues with updates from the remaining clients. A single node's failure does not block global progress.

# I  ANALYSIS

In this section, we present the theoretical analysis of an one-step local gradient update, comparing ORTHOFL which calibrates local model weights, with conventional asynchronous methods (e.g., FedAsync) which directly assign the aggregated global weight to the client model. We characterize the conditions under which ORTHOFL yields better updates than conventional methods.

**Notation and setup**. Fix a client $m$ with local objective $F_m : \mathbb{R}^d \to \mathbb{R}$. Let $L_m > 0$ be a smoothness constant: for all $X, Y$, $\|\nabla F_m(X) - \nabla F_m(Y)\| \leq L_m \|X - Y\|$ (equivalently, $F_m$ is $L_m$-smooth). At global round $t$, denote the current client model weight $W_m^{(t)}$ and the current global state $W^{(t)}$. Let $\tau < t$ be the previous global round when client $m$ communicated with the server. Define $g_m$ as the gradient of $F_m$ at $W_m^{(t)}$:

$$g_m := \nabla F_m\left(W_m^{(t)}\right), \qquad G_m := \|g_m\|.$$

Define the weight changes for the global model and the client model $m$ during delay $t - \tau$ as:

$$\Delta := W^{(t)} - W^{(\tau+)}, \qquad \Delta_m := W_m^{(t)} - W_m^{(\tau+)}.$$

Set $u := \Delta_m / \|\Delta_m\|$ as the unit vector pointing in the direction of local client update. Let $\Delta\alpha := \|\Delta_m\|$. Decompose the global shift into the $u$-axis and its orthogonal complement $\delta$:

$$\Delta = \alpha u + \delta, \qquad \langle \delta, u \rangle = 0, \qquad \alpha := \langle \Delta, u \rangle.$$

The difference between the global state and the client model weight at $\tau_+$ can be represented as:

$$D := W^{(\tau+)} - W_m^{(\tau+)} = \rho u + d, \qquad \langle d, u \rangle = 0.$$

Let $S_{\text{ORTHOFL}}$ and $S_{\text{baseline}}$ denote the starting points of the client model before the one-step local gradient update in ORTHOFL and in the baseline, respectively. Let $s_{\text{ORTHOFL}}$ and $s_{\text{baseline}}$ denote the gaps between these starting points and the client's local state at the end of its previous training. They are defined as follows:

- **ORTHOFL.** The client adds the orthogonal part of the global shift to its weight, so it starts from:

$$S_{\text{ORTHOFL}} := W_m^{(t)} + \delta, \qquad s_{\text{ORTHOFL}} := S_{\text{ORTHOFL}} - W_m^{(t)} = \delta.$$

- **Baseline (directly assigning global weights to clients).** The server aggregates the global model by moving average: $W^{(t+)} = (1 - \beta_t)W^{(t)} + \beta_t W_m^{(t)}$, where $\beta_t \in [0, 1]$ is the global moving-average weight at round $t$. The client starts from this aggregated global weight:

$$S_{\text{baseline}} := W^{(t+)} = W_m^{(t)} + (1 - \beta_t)\big((\rho + \alpha - \Delta\alpha)u + (d + \delta)\big),$$

so

$$s_{\text{baseline}} := S_{\text{baseline}} - W_m^{(t)} = (1 - \beta_t)\big((\rho + \alpha - \Delta\alpha)u + (d + \delta)\big).$$

For a step size $\eta > 0$, we define the difference of loss of ORTHOFL and the baseline after one step update as:

$$\Delta_{1\text{step}} := F_m\big(S_{\text{baseline}} - \eta\nabla F_m(S_{\text{baseline}})\big) - F_m\big(S_{\text{ORTHOFL}} - \eta\nabla F_m(S_{\text{ORTHOFL}})\big).$$

**Lemma I.1** (One-step bounds for $L_m$-smooth $F_m$). *Let $F_m : \mathbb{R}^d \to \mathbb{R}$ have $L_m$-Lipschitz gradient, i.e.,*

$$\|\nabla F_m(x) - \nabla F_m(y)\| \le L_m \|x - y\| \qquad \forall x, y \in \mathbb{R}^d.$$

*Then for any point $S \in \mathbb{R}^d$ and any step size $\eta \ge 0$,*

$$F_m\big(S - \eta\nabla F_m(S)\big) \le F_m(S) - \eta\Big(1 - \tfrac{L_m\eta}{2}\Big)\|\nabla F_m(S)\|^2, \tag{12}$$

$$F_m\big(S - \eta\nabla F_m(S)\big) \ge F_m(S) - \eta\Big(1 + \tfrac{L_m\eta}{2}\Big)\|\nabla F_m(S)\|^2. \tag{13}$$

*Equivalently, we have*

$$\Big| F_m\big(S - \eta\nabla F_m(S)\big) - F_m(S) + \eta\|\nabla F_m(S)\|^2 \Big| \le \tfrac{L_m}{2}\eta^2 \|\nabla F_m(S)\|^2. \tag{14}$$

*In particular, if $0 < \eta \le 1/L_m$, then*

$$F_m\big(S - \eta\nabla F_m(S)\big) \le F_m(S) - \tfrac{\eta}{2}\|\nabla F_m(S)\|^2.$$

*Proof.* The $L_m$-smoothness (Lipschitz gradient) condition is equivalent to the standard upper bound

$$F_m(y) \le F_m(x) + \langle\nabla F_m(x),\, y - x\rangle + \tfrac{L_m}{2}\|y - x\|^2 \qquad \forall x, y, \tag{15}$$

and the corresponding two-sided form

$$\big| F_m(y) - F_m(x) - \langle\nabla F_m(x),\, y - x\rangle \big| \le \tfrac{L_m}{2}\|y - x\|^2. \tag{16}$$

Set $x = S$ and $y = S - \eta\nabla F_m(S)$, so $y - x = -\eta\nabla F_m(S)$. Then

$$F_m(y) - F_m(x) + \eta\|\nabla F_m(S)\|^2 = F_m(y) - F_m(x) - \langle\nabla F_m(S),\, y - x\rangle.$$

Applying (16) yields (14). Expanding (14) to the "$\le$" and "$\ge$" directions gives (12) and (13), respectively. $\qquad\square$

**Lemma I.2** (Pre-step lower bound). *With the notation above, for any client $m$,*

$$\Delta_{\text{pre}} := F_m(S_{baseline}) - F_m(S_{\text{ORTHOFL}})$$

$$\ge -(1 - \beta_t)\Big(|\rho + \alpha - \Delta\alpha|\,|\langle g_m, u\rangle| + G_m\|d\|\Big) - \beta_t G_m\|\delta\|$$

$$- \tfrac{L_m}{2}\Big((1 - \beta_t)^2\big((\rho + \alpha - \Delta\alpha)^2 + \|d + \delta\|^2\big) + \|\delta\|^2\Big). \tag{17}$$

*Proof.* Apply the two-sided first-order bound:

$$\big| F_m(X + s) - F_m(X) - \langle\nabla F_m(X),\, s\rangle \big| \le \tfrac{L_m}{2}\|s\|^2.$$

Using base $X = W_m^{(t)}$ and steps $s_{\text{baseline}}$ and $s_{\text{ORTHOFL}} = \delta$, subtract the two inequalities to get

$$\Delta_{\text{pre}} \ge \langle g_m,\, s_{\text{baseline}} - s_{\text{ORTHOFL}}\rangle - \tfrac{L_m}{2}\big(\|s_{\text{baseline}}\|^2 + \|s_{\text{ORTHOFL}}\|^2\big).$$

Since $s_{\text{baseline}} - s_{\text{ORTHOFL}} = (1 - \beta_t)(\rho + \alpha - \Delta\alpha)u + (1 - \beta_t)d - \beta_t\delta$, Cauchy–Schwarz yields $\langle g_m,\, s_{\text{baseline}} - s_{\text{ORTHOFL}}\rangle \ge -(1 - \beta_t)|\rho + \alpha - \Delta\alpha|\,|\langle g_m, u\rangle| - (1 - \beta_t)G_m\|d\| - \beta_t G_m\|\delta\|$, and $\|s_{\text{baseline}}\|^2 = (1 - \beta_t)^2\big((\rho + \alpha - \Delta\alpha)^2 + \|d + \delta\|^2\big)$, $\|s_{\text{ORTHOFL}}\|^2 = \|\delta\|^2$. Combine to obtain (17). $\qquad\square$

**Proposition 1** (One-step sufficient condition). *For any step size $\eta \in (0, 1/L_m]$,*

$$\Delta_{1\text{step}} \ge \Delta_{\text{pre}} - \eta\Big(1 + \tfrac{L_m\eta}{2}\Big)\|\nabla F_m(S_{baseline})\|^2 + \eta\Big(1 - \tfrac{L_m\eta}{2}\Big)\|\nabla F_m(S_{\text{ORTHOFL}})\|^2. \tag{18}$$

*Consequently, using Lipschitzness of $\nabla F_m$,*

$$\|\nabla F_m(S_{baseline})\| \le G_m + L_m \|s_{baseline}\|, \qquad \|\nabla F_m(S_{\text{ORTHOFL}})\| \ge \max\{0,\, G_m - L_m \|\delta\|\},$$

*a* sufficient *condition for $\Delta_{1\text{step}} \ge 0$ is:*

$$
\begin{aligned}
&(1-\beta_t)\,|\rho + \alpha - \Delta\alpha|\,|\langle g_m,\, u\rangle| \;+\; (1-\beta_t)G_m \|d\| \;+\; \beta_t G_m \|\delta\| \\
&\quad + \tfrac{L_m}{2}\Big((1-\beta_t)^2\big((\rho+\alpha-\Delta\alpha)^2 + \|d+\delta\|^2\big) + \|\delta\|^2\Big) \\
&\le \eta\Big(1 - \tfrac{L_m\eta}{2}\Big)\big(\max\{0,\, G_m - L_m\|\delta\|\}\big)^2 \;-\; \eta\Big(1 + \tfrac{L_m\eta}{2}\Big)\big(G_m + L_m\|s_{baseline}\|\big)^2.
\end{aligned}
\tag{19}
$$

*In particular, if* (19) *holds for some $\eta \in (0, 1/L_m]$, then*

$$F_m\big(S_{\text{ORTHOFL}} - \eta\nabla F_m(S_{\text{ORTHOFL}})\big) \;\le\; F_m\big(S_{baseline} - \eta\nabla F_m(S_{baseline})\big).$$

*Proof.* The one-step bounds for $L_m$-smooth $F_m$ gives, for any $S$ and $\eta > 0$,

$$F_m(S - \eta\nabla F_m(S)) \le F_m(S) - \eta\Big(1 - \tfrac{L_m\eta}{2}\Big)\|\nabla F_m(S)\|^2,$$

and also the reverse inequality $F_m(S - \eta\nabla F_m(S)) \ge F_m(S) - \eta\Big(1 + \tfrac{L_m\eta}{2}\Big)\|\nabla F_m(S)\|^2$. Apply a lower bound to the baseline and an upper bound to ORTHOFL, then subtract to obtain (18). Bound the gradients by Lipschitzness around $W_m^{(t)}$ to get (19). $\square$

