# OpenReview forum: "Orthogonal Calibration for Asynchronous Federated Learning"
_ICLR.cc/2026/Conference — Submitted to ICLR 2026_

### Official Review · Reviewer_voAj · 2025-10-28

**Soundness:** 3
**Presentation:** 4
**Contribution:** 2
**Rating:** 2
**Confidence:** 3

**Summary:**

The paper targets the challenges of potential gradient conflict in AFL due to the asynchronous updates between the server and clients. The authors propose to decouple global and local learning progress by orthogonalizing their weights (per client) before applying it to client weights. Experiments show higher accuracy and substantial speed-ups compared to synchronous and consistently outperform several asynchronous baselines across various datasets.

**Strengths:**

The authors propose an interesting perspective of looking at the client drift in AFL as from "spatial drift" and "temporal staleness", though the concept has always been there.
The paper is well written with clear motivation and algorithmic description.
The results show solid empirical improvements on various datasets against both synchronized FL and AFL.

**Weaknesses:**

The major ideas, staleness-aware and orthogonal learning have both been quite well explored, relevant works can be easily found in papers published in recent years. Therefore, the novelty and originality become quite limited.

The convergence rate has only been shown with empirical results without math analysis.

For larger models, the operation of orthogonalization may raise costs worth considering. This is overlooked in the draft.

Removing the projection may not help reduce bias, it may sometimes even favor the frequent clients resulting in larger bias.

**Questions:**

1. Please clarify the distinction between this work and related works that focus on staleness-awareness and orthogonal learning in AFL and in FL, because some orthogonal learning approaches in synchronized setup can be applied to AFL as well.

2. Please analyze the global convergence rate.

3. Could the authors discuss the orthognalization operation's cost?

4. What do the authors think about the potential bias problem? Could orthognoal help it? How?

---

> ### Author Response · Authors · 2025-11-21
> **Response to Reviewer voAj**
>
> We thank the reviewer for the comments. Below, we address each question and clarify points that may be overlooked.
>
> 1. **Distinction from prior works:** Orthogonal learning in centralized settings was discussed in Section F. There are fundamentally different optimization constraints because AFL must integrate client updates without accessing data for privacy reasons, while centralized approaches can revisit data or sample-level gradients (Lines 686–690). In addition, orthogonal learning works in synchronous FL [1–4] rely on the below properties:
> - All clients update from the same global model each round, so their updates diverge only spatially (due to non-IID data) around a shared global reference.
> - Orthogonalization is performed among client updates in a single, consistent coordinate frame.
>
> In AFL, neither assumption holds in AFL. Clients start from different global snapshots, and the global model evolves while they train. As a result, there is no shared coordinate frame across clients, and their updates are not mutually comparable. To address this fundamental difference, OrthoFL reformulates the orthogonalization objective, which decouples global and local learning progress and orthogonalizes global model drift against the client update. We will add this clarification to the paper.
>
> [1] Federated Orthogonal Training: Mitigating Global Catastrophic Forgetting in Continual Federated Learning
>
> [2] FediOS: Decoupling Orthogonal Subspaces for Personalization in Feature-skew Federated Learning
>
> [3] FedSOL: Stabilized Orthogonal Learning with Proximal Restrictions in Federated Learning
>
> [4] Maximum Knowledge Orthogonality Reconstruction with Gradients in Federated Learning
>
> 2. **Global convergence analysis:** Deriving a global convergence rate is challenging because non-convexity, non-IID data, and asynchronous stale updates in our setting make long-term error accumulation difficult to control. Existing theoretical tools generally require additional assumptions (e.g., IID, bounded staleness, an artificial synchronized timeline, simplified update rules, convex loss) that are unrealistic for practical AFL and are therefore not directly applicable to OrthoFL. In this context, our work follows the precedent of many recent FL works that focus on strong experimental validation. Empirically, OrthoFL converges reliably and achieves the fastest convergence rate among all baselines (See Figure 9).
>
> 3. **Orthogonalization cost:** The orthogonalization overhead was already discussed in Section 5.6. For example, on MNIST, orthogonalization adds only 0.003 seconds per round, whereas communication and client training take ~3 mins. The orthogonalization cost (O(P), P is the number of model parameters) is minimal compared with client training (O(EBP), E is the number of local epochs and B is the number of mini-batches) (Line 320 - 321). As long as a client can perform standard local training, it can easily afford orthogonalization.
>
> 4. **Potential bias issue:** OrthoFL helps mitigate bias. Its calibration removes interference often introduced by frequent clients whose updates dominate the global trajectory. The added component is orthogonal to the client's own optimization direction, so the client's learning is less affected by bias from other clients.

---

### Official Review · Reviewer_hVGm · 2025-10-30

**Soundness:** 3
**Presentation:** 2
**Contribution:** 2
**Rating:** 2
**Confidence:** 3

**Summary:**

Can the authors provide any theoretical justification for LEAN's convergence and stability?
How sensitive is the method to the choice of library size (n) and checkout size (m), and how should they be set?
What is the inference cost of the final summed-pair adapter compared to a standard LoRA adapter?
Have the authors considered more intelligent strategies for sampling basis pairs from the library (e.g., based on staleness)?

**Strengths:**

The use of orthogonal projection to decouple conflicting updates is an elegant, geometrically intuitive, and novel solution. The evaluation uses strong baselines, diverse tasks, and realistic delay simulations to convincingly demonstrate the method's robustness and superiority. The evaluation is a major strength, using strong baselines, diverse tasks, and realistic delay simulations to convincingly demonstrate the method's robustness and superiority.

**Weaknesses:**

1. The theoretical analysis is limited. The analysis in Appendix I is a one-step analysis, showing that under certain conditions, a single step of ORTHOFL is better than a baseline. This is insightful but does not constitute a full convergence proof over the entire training process. For a top-tier conference like ICLR, a more complete theoretical treatment guaranteeing convergence would significantly strengthen the paper.
2. The ORTHOFL algorithm, particularly the server-side implementation (Algorithm 1), requires the server to store the historical state W(τ+) for each client currently performing local training. In a large-scale system with thousands or millions of clients, where many could be "in-flight" at any given time, the memory overhead of storing a full model copy for each active client could become a significant scalability bottleneck. While the client-side version (Algorithm 2) mitigates this, it places more burden on the client, and the paper could discuss this trade-off more explicitly.
3. The method projects the global shift ∆ onto the client's update direction ∆m. ∆m is defined as W(t)m - W(τ+)m, which represents the total change in the client model over E local epochs. This vector is an aggregation of many local gradient steps. It's not immediately obvious that this single aggregate direction is the most representative one to project against.

**Questions:**

1. What are the main challenges in extending the one-step analysis to a full convergence proof?
2. How do you address the potential server-side memory bottleneck from storing historical states for many concurrent clients?
3. Why does layer-wise projection outperform full-model projection? Does this imply drift is a layer-specific issue?
4. Have you experimented with projecting against other directions besides the total client update vector ∆m?

---

> ### Author Response · Authors · 2025-11-21
> **Response to Reviewer hVGm**
>
> **Challenge in full convergence proof:** The main challenges are from the combination of non-convexity, non-IID data, and the complex asynchronous update dynamics in our setting. These factors make it hard to control the long-term error accumulation. Existing theoretical tools generally require additional assumptions (e.g., IID, bounded staleness, an artificial synchronized timeline, simplified update rules, convex loss) that are unrealistic for practical AFL and are therefore not directly applicable to OrthoFL. In this context, our work follows the precedent of many recent FL works that focus on strong experimental validation. Empirically, OrthoFL converges reliably and achieves the fastest convergence rate among all baselines (See Figure 9).
>
> **Server- vs client-side algorithm choice:** We clarify that the choice between server- or client-side orthogonalization is primarily a protocol preference with little practical trade-off. In both cases, orthogonalization overhead is affordable. Orthogonalization involves only basic vector operations (dot products and subtraction), so **any client capable of running standard local SGD can easily handle orthogonalization**. The overhead was evaluated in Section 5.6. For example, on MNIST, orthogonalization adds only 0.003 seconds per round, whereas communication and client training take roughly 3 minutes. Therefore, when the server faces storage bottlenecks, switching to the client-side version is straightforward.
>
> **Advantage of the layer-wise projection:** As explained in Lines 236–241, layer-wise projection approximates a block-diagonal projection that respects the distinct functional roles of different layers (e.g., in CNNs, early layers capturing low-level features vs. deeper layers encoding high-level semantics) or different scales. A full-model projection flattens all parameters into one vector, mixing incompatible subspaces. Furthermore, when projected together, layers with much larger or smaller magnitudes disproportionately influence the projection and distort updates in unrelated layers. Our ablation (Figure 3) OrthoFL-Full Proj. validates the full-model projection performs worse. The improved performance of layer-wise projection does not imply that drift originates at the layer level. Drift is a global effect of staleness. Layer-wise projection prevents cross-layer interference by respecting the model's architectural structure.
>
> **Projection against other directions:** In Section 5.3, we evaluated another orthogonalization strategy OrthoFL-Pairwise Proj., which projects ∆m onto the orthogonal subspace of the most recent updates from all other clients. This achieves performance comparable to using the aggregated global change ∆, showing that multiple orthogonalization choices can be effective (Lines 368). However, such alternatives require storing or transmitting substantially more state and introduce higher overhead under FL privacy constraints. In contrast, using the aggregated update ∆ provides a simple, lightweight, and privacy-compliant approximation that is both effective and efficient.
> Our goal is not to find the most representative direction in an *absolute* sense, but to identify one that has strong performance under FL constraints without additional privacy risk. As noted in Lines 369–370, exploring more advanced orthogonalization strategies is a promising future work.

---

### Official Review · Reviewer_aHJa · 2025-10-31

**Soundness:** 3
**Presentation:** 2
**Contribution:** 2
**Rating:** 4
**Confidence:** 4

**Summary:**

This paper proposes the ORTHOFL method to decouple global and local learning progress, thereby reducing model drift. Additionally, this approach optimizes weights on both client and server sides to enhance accuracy. Based on these methods, the authors present some new result on asynchronous federated learning.

**Strengths:**

1. The experiments in this paper are conducted comprehensively, though some minor flaws exist in certain details.

2. Most methods mentioned in the paper are elaborated upon in detail within the appendix.

3. By integrating the experiments in the main text with the experimental results in the appendix, it is evident that the proposed method in this paper demonstrates certain generalization capabilities and cutting-edge potential.

**Weaknesses:**

1. The author mentions improving model performance by optimizing weights on both client and server sides. It is suggested to add 1-2 adaptive weighting methods to the comparative experiments to demonstrate the model's advantages.

2. Table 2 presents the performance of the proposed model and comparison models across five datasets. However, the subsequent ablation experiments only include three datasets. It is recommended to include the remaining two datasets here.

3. The experimental sections in Figures 4 and 5 also utilize two or three datasets. It is recommended to explain why only these datasets were selected. Alternatively, all datasets should be included as in the previous benchmark experiments.

4. Figures and tables are not consistently referenced in the text. For example, Figure 3 is not referenced within the body.

5. The formatting of references appears inconsistent. It is recommended to thoroughly proofread them according to the conference's formatting requirements.

6. The text contains numerous long, complex sentences separated by commas, along with some fundamental grammatical issues. It is recommended that the author carefully proofread the entire document to enhance the reader's experience.

7. It is recommended to include an overall model architecture diagram to help readers understand the model's structure.

**Questions:**

1. Will the orthogonal direction gradually deviate from the global optimum? Is there bounded deviation?

2.Does orthogonal calibration guarantee long-term convergence?

3.Beyond the advantages outlined in the paper, has consideration been given to risks such as privacy breaches? Have the authors implemented corresponding measures to address these risks?

---

> ### Author Response · Authors · 2025-11-21
> **Response to Reviewer aHJa**
>
> We thank the reviewer for the comments. Our responses to the questions are provided below.
> 1. **Will orthogonal direction gradually deviate from the global optimum**: No. OrthoFL does not calibrate the global model, but instead calibrates the local initialization before each client's training. The client then performs local optimization, which further corrects the update toward its objective. Therefore, it is a local adjustment and does not introduce systematic deviation to global optimum.
>
> 2. **Long-term convergence:** Empirically, OrthoFL converges reliably and achieves the fastest convergence rate among all baselines (See Figure 9). From a theoretical perspective, long-term convergence analysis remains challenging due to non-convexity, non-IID data, and the complex asynchronous update dynamics in our setting. Existing theoretical tools generally require additional assumptions (e.g., IID, bounded staleness, an artificial synchronized timeline, simplified update rules, convex loss) that are unrealistic for practical AFL and are therefore not directly applicable to OrthoFL. In this context, our work follows the precedent of many recent empirical works in federated learning that focus on robust design and strong experimental validation. We acknowledge the value of long-term convergence analysis and view it as an interesting direction for future work, but not essential to validate our contributions.
>
> 3. **Privacy considerations:** ORTHOFL introduces no new privacy risks beyond standard federated learning. Same as typical FL, only model weights are shared, and OrthoFL is fully compatible with established differential privacy (DP) techniques. Clients can independently apply DP measures, such as gradient clipping and adding noise, before sending updates. Thus, OrthoFL does not weaken existing privacy guarantees and can be paired with existing DP methods for FL if desirable. Given that this work focuses on asynchronous optimization, detailed privacy mechanisms are left for future work, and we will add this clarification to the paper.

---

### Official Review · Reviewer_UdwP · 2025-10-31

**Soundness:** 3
**Presentation:** 3
**Contribution:** 3
**Rating:** 6
**Confidence:** 4

**Summary:**

This paper introduces ORTHOFL, an asynchronous federated learning framework designed to mitigate interference between global model progress and local client training. The core idea is to decouple the global and local model weights. When a client's update arrives, the server first aggregates it into the global model using a staleness-aware moving average. Then, to update the client's model for its next training round, it calculates the global model's shift during the client's delay period and removes the component of this shift that is parallel to the client's own update direction. This is done via a layer-wise orthogonal projection. The resulting "calibrated" global shift is then sent back to the client. Experiments on diverse datasets and under various delay simulations show that ORTHOFL improves accuracy and convergence speed compared to synchronous and other asynchronous baselines.

**Strengths:**

* The core idea of using orthogonal calibration to decouple and manage the interference between global and local updates is novel and intuitive.
* The paper is well-written and clearly presented, making the motivation and methodology easy to follow.
* The experiments are comprehensive, covering multiple datasets, a strong set of synchronous and asynchronous baselines, and various simulated delay distributions.

**Weaknesses:**

* The paper claims layer-wise projection is "memory-efficient" in Section 4.1. However, storing the model weights for projection, whether layer-by-layer or as a single flattened vector, seems to require the same amount of storage space. Could the authors clarify this claim?
* The method excludes the component of the global shift that is projected onto the local update. Should this information always be discarded? Since local models can be biased due to non-IID data, this projected component (whether aligned or opposing) might contain useful information to either reinforce or correct the local update direction.
* The ablation study shows that removing the global moving average (ORTHOFL w/o MA) hurts performance more than removing calibration (ORTHOFL w/o Calib.). This suggests that modifying the global aggregation contributes more to the performance gain than the orthogonal calibration itself. Does this finding support the concern that excluding the projected component might be discarding useful information?
* Figure 1 effectively visualizes the oscillating global weight changes to motivate the problem. It would be very helpful to include a similar plot for ORTHOFL. A direct comparison would make it easier to understand the effect of the proposed calibration method.

**Questions:**

* How does the magnitude of the calibrated global shift, compare to the magnitude of the original global shift? Does this relationship change as training progresses? As these indirectly influence the local step size for updating the local parameter in Eq.(7).

* Following the above, the global aggregation weight βt in Eq. (3) currently only depends on staleness. The corresponding adjustment of global aggregation weights of different clients also remains a question. Have the authors considered adjusting βt based on information from the projection?

---

> ### Author Response · Authors · 2025-11-21
> **Response to Reviewer UdwP**
>
> We thank the reviewer for the constructive comments and insightful questions. Our responses are provided below.
>
> **Memory-efficeint:** "Memory-efficient" refers to computational memory, not storage. Layer-wise projection avoids constructing a large flattened vector and operating on it as a single block, which reduces peak temporary memory usage during projection. We will clarify this in the paper.
>
> **Is projected-away component discarding useful information?** The aligned component is removed not because it is harmful, but because it is redundant with what the client already captures during local training (Line 232-233). OrthoFL adjusts the local *initialization*, after which the client performs training that naturally reinforces or corrects movement along its own update direction. Keeping the projected component would double-count movement along that direction and amplify bias under non-IID data. ORTHOFL therefore preserves only the part that provides new information (orthogonal to the client's progress) and relies on local training to handle the component aligned with the client's direction.
>
> **Ablation study of removing the global moving average:** The ablation OrthoFL w/o MA overwrites the global model with the calibrated client model. Because the calibration adds the global shift to the client's weight (rather than adding the client update to the global weight), it is biased toward the client’s local distribution and performs worse on the global objective. This is a more aggressive change than removing calibration, and it highlights the importance of decoupling global aggregation from local learning in AFL.
>
> **Magnitude of calibrated vs. original global shift:** We observe the difference between the magnitude of the original global shift and the calibrated shift (‖Δ‖ − ‖Δ⊥‖) is relatively large during the first few updates for each client (about the first 20% of training), and it then becomes small as training progresses. This is expected as once the early conflicting component is removed, update directions in a high-dimensional, non-IID setting tend to have little overlap as the model stabilizes.
>
> Following the reviewer's suggestion, we tested whether this difference could be used to adjust the aggregation weight βt in Eq. (3). Specifically, we set: β_t​=β⋅(100⋅(‖Δ‖-‖Δ⊥‖)+1)^{−a}. This quantity reflects the extent of interference during client's delay and could indicate staleness and alignment. The table below shows the results.
>
> | Method                            | MNIST Acc     | MNIST Time | CIFAR10 Acc | CIFAR10 Time | HAR Acc| HAR Time |
> |------------------------------|-----------------|------------|------------------|--------------|------------------|----------|
> | OrthoFL                           | 98.2 ± 0.3       | 0.18×      | 76.5 ± 3.3       | 0.19×        | 89.7 ± 1.0       | 0.23×    |
> | OrthoFL (proj-aware β_t) | 97.2 ± 0.7       | 0.22×      | 74.15 ± 4.3     | 0.25×        | 88.5 ± 1.9       | 0.20×    |
>
> The results show that the projection-aware β_t underperforms the original staleness-based β_t, suggesting this magnitude difference does not reliably correlate with how much a delayed update should contribute to the global model.

---

### Meta-Review · Area_Chair_5aU7 · 2025-12-16

**Summary:**

Most of the reviewers agree that this paper tackles a practical issue in asynchronous FL, i.e., the interference between delayed global updates and ongoing client-local updates. To address this issue, this paper proposes a geometric fix via orthogonal calibration/projection to reduce destructive coupling. The performance of the proposed OrthoFL is validated via experiments across multiple datasets.

The authors provide rebuttals to each reviewer's comments or concerns. Here are a summary of the major concerns raised for this work:
- One reviewer views the core idea as incremental and potentially overlapping with existing orthogonalization and projection approaches in FL. Despite the rebuttal, it is still not convincing on the technical novelty positioned over existing methods. It would be beneficial to this paper via clearer distinctions over current methods.
- The proposed framework is purely heuristic without any theoretical justifications. Multiple reviewers note the lack of strong theoretical grounding, e.g., convergence, bias, under realistic asynchronous setting with non-IID dynamics. As a result, some assumptions for analysis may be hard to justify.
- There are several key questions on the method details and the ablations. In particular, the reviewers question about what is "discarded"by orthogonalization, what is the sensitivity of OrthoFL to hyperparameters such as projection strength, and whether additional ablations are needed to pinpoint when and why OrthoFL helps. In addition, there are also concerns on the practicality of the proposed OrthoFL in terms of the server-side overhead and whether OrthoFL scales cleanly in large deployments.

**Reviewer Concerns:**

Concerns partially addressed by the rebuttal
- The authors clarify memory and compute costs and argue that the projection is lightweight, and that OrthoFL can be deployed with minimal added server overhead relative to existing async aggregation.
- The rebuttal also provides an explanation framed around "spatial drift" vs. "temporal drift," and how calibrating updates in an orthogonal subspace reduces interference between stale global directions and current local descent.
- The authors also respond to requests on what is being projected, why projection should not destroy useful signal in practice, and provide additional discussion or clarity around observed improvements and failure modes in the experiments.

Concerns still outstanding:
- Even with added citations and discussion, the most negative reviewer’s core point on "this resembles known orthogonalization/projection tricks and lacks a clear new conceptual contribution" is only partially mitigated unless the paper very explicitly delineates including but not limited to what is new, e.g., from the perspectives of calibration target, where projection is applied, interaction with async staleness, what prior methods cannot do, and why so.
- The rebuttal argues that full theory is challenging in non-convex async FL. Unfortunately, most of the reviewers cannot agree on this point as there are extensive works on convergence analysis for non-convex FL or DFL both in synchronous and asynchronous settings. Even without strong convergence guarantees, at least a principled characterization may be useful along with empirical validations.
- One reviewer explicitly worries that projecting out components could remove helpful information and asks for deeper ablations showing when this does or does not happen. Although the rebuttal explains empirically, this concern has not been fully addressed and lacks some performance analysis, e.g., performance vs. projection rate, gradient alignment statistics, or cases where projection hurts.

**Reviewer Scores:**

Reviewer UdwP is already positive and focuses on clarity, ablations and projection details. The rebuttal addresses mechanism and memory/overhead questions.

Reviewer aHJa's concerns are mainly empirical completeness and clarity. The rebuttal provides clarifications and positions privacy as future work. If we assume that the promised experimental clarifications are convincing, the reviewer might consider raise the score.

Reviewer hVGm and Reviewer voAj's critiques are fundamentally about novelty and missing theory, plus concerns about bias/cost and limited differentiation from prior work. The rebuttal helps on overhead and positioning, but unless the novelty is made unambiguously distinct, and/or a theoretical result is added, they likely remain negative.

---

### Decision · Program_Chairs · 2026-01-26

Reject